# Comparative Study on Hydrolysis, Physicochemical and Antioxidant Properties in Simulated Digestion System between Cooked Pork and Fish Meat

**DOI:** 10.3390/foods12091757

**Published:** 2023-04-23

**Authors:** Yuhan Chen, Hanzhi Jing, Shanbai Xiong, Anne Manyande, Hongying Du

**Affiliations:** 1Key Laboratory of Environment Correlative Dietology, College of Food Science and Technology, Huazhong Agricultural University, Ministry of Education, Wuhan 430070, China; 2School of Human and Social Sciences, University of West London, Middlesex TW8 9GA, UK; 3Department of Food Science and Engineering, College of Light Industry and Food Engineering, Nanjing Forestry University, Nanjing 210037, China

**Keywords:** pork, grass carp, digestion, hydrolysis, antioxidant properties, protein structure

## Abstract

Pork and grass carp are commonly consumed animal protein sources, classified as red meat and white meat, respectively. This study aimed to better understand the differences in digestive behavior, nutrition, and functionality during digestion between these two types of meat after fat removal. The results showed that grass carp was more easily digested than pork, with a higher degree of hydrolysis, a smaller protein particle size, and a greater release of oligopeptides and amino acids (*p* < 0.05). During gastric digestion, all α-helix structures were destroyed, and the effect of the whole digestion process on the secondary and tertiary structure of pork protein was greater than that of grass carp. The antioxidant properties of the digestive fluids from the two types of meat showed different strengths in various assays, but the correlation analysis revealed that TCA-soluble peptides, random coil content, and particle size significantly influenced both types of meat. These findings provide new insights into the structural state and antioxidant properties of protein in meat digestion, which contribute to our understanding of the nutritional value of pork and grass carp.

## 1. Introduction

As a high-quality source of protein, meat plays an indispensable role in the typical human diet. Generally, white meat is considered to be healthier than red meat. This is because red meat is typically high in fat, especially saturated fatty acids. These types of fat are thought to be the primary cause of elevated blood cholesterol levels, triacylglycerols, and low-density lipoprotein cholesterol (LDL-C), which can increase the risk of coronary heart disease, hypertension, and type II diabetes [1,2,3]; some studies suggest that the exogenous sialic acid in red meat may cause an inflammatory response in cells, which could increase the likelihood of cellular carcinogenesis. However, other studies have found that the issue may not be in the muscle itself [4]. Instead, the alleged carcinogenicity of red meat may be due to inappropriate cooking practices that result in the production of heterocyclic aromatic hydrocarbons and heterocyclic amines, which are not exclusive to red meat [5]. According to these studies, red meat with the fat removed may not result in any significant difference in terms of health when compared with white meat. Despite being classified as red meat, pork, which is still one of the most widely consumed meats in the world, is considered irreplaceable from both economic and cultural perspectives for a significant portion of the population [6,7]. Given that the harmful effects of pork muscle have not been established, exploring the digestive, nutritional, and functional differences between pork and certain white meats is a worthwhile endeavor.

Grass carp, a type of white meat, is highly valued for its tender flesh and relatively low farming costs. According to the Food and Agriculture Organization (FAO) in 2022, grass carp was the largest fish species in the world in terms of total production in 2020; as a source of protein and income for rural communities in low-income countries, grass carp is becoming an important food security and poverty alleviation tool for developing nations [8,9]. Several studies have been conducted on the flavor substances [10], functional properties of hydrolysates [11,12], and physicochemical properties of fish skin collagen [13] in grass carp as it is a typical white meat. However, there is still a need for more research on the digestibility and digestates of grass carp meat.

In vitro digestion (IVD) is a useful model for evaluating the digestive performance of food as an alternative to complex in vivo systems [14]. With the emergence of INFOGEST, an international standard method for IVD, the convenience and generality of this model have been amplified [15]. INFOGEST provides recommendations for various digestion conditions, including digestive enzyme activity, pH, temperature, and digestive fluid salt concentration. It encourages researchers to conduct digestion experiments under internationally agreed conditions, while allowing for some flexibility to adjust certain conditions based on specific situations [16].

After digestion in the gastrointestinal tract, some high-quality proteins can be degraded into bioactive peptides with certain functionalities for the human body, such as antithrombotic peptides, anti-inflammatory peptides, antioxidant peptides, and antibacterial peptides, among others [17]. The proceeding digestion profoundly affects the properties of these peptides. It has been found that antioxidant peptides are often present in the digestion of meat proteins [18], and their activity is exposed by the hydrolysis of digestive enzymes. However, further digestion may lead to the weakening or loss of their activity [19]. Therefore, the antioxidant activity of the products obtained after meat protein digestion is likely to be influenced by multiple factors such as protein source, degree of digestion, and amino acid ratios. The aim of this study was to analyze and compare the digestive, nutritional, and functional differences between pork and fish using the IVD model, and to determine any intrinsic link between them through a correlation analysis of some of these data.

## 2. Materials and Methods

### 2.1. Materials

The white pig longissimus dorsi muscle and grass carp back muscle were purchased from Zhongbai supermarket at Huazhong Agricultural University. Pepsin (P7012, from porcine gastric mucosa, 3344 U/mg) and trypsin (T7409, from porcine pancreas, 1369 U/mg) were obtained from Sigma Aldrich (St Louis, MO, USA). BCA protein assay kit (P0012), Total Antioxidant Capacity Assay Kit with ABTS method (S0119), and FRAP method (S0116) were sourced from Beyotime Biotech Co., Ltd. (Shanghai, China). The 12% precast gels were obtained from Bio-Rad (Irvine, CA, USA). All other chemicals used in the study were analytical grade and acquired from Aladdin Biotech Co., Ltd. (Shanghai, China).

### 2.2. Meat Cooking

Firstly, the visible fat, connective tissue, and bones were carefully removed from pork and grass carp muscles. Then, the muscles were grated separately with a meat grinder. Similar to a former protocol [20], every sample was packed in a retort pouch and immersed in a 72 °C water bath until the central temperature reached 70 °C.

### 2.3. Preparation of IVD

The IVD method for cooked meat was obtained with certain modifications based on INFOGEST [15]. Briefly, during the oral phase, 5 g cooked meat mixed with 5.0 mL of simulated salivary fluid (SSF) was incubated with vibration for 2 min in a 37 °C water bath. In the gastric phase, the oral chyme was mixed with simulated gastric fluid (SGF) containing pepsin (2000 U/mL) to 20.0 mL, and pH was adjusted to 3.0 with 1 M HCl. The mixture was kept at 37 °C with continual shaking for 120 min. For the intestinal phase, the gastric chyme was mixed with simulated intestinal fluid (SIF) containing trypsin (100 U/mL) to 40.0 mL, and pH was adjusted to 7.0 with 1 M NaOH. The mixture was kept at the same conditions as above. The formulation of salivary, gastric, and intestinal fluids is shown in Figure 1.

During sampling, NaOH was added to the gastric phase to pH = 8.0 to end the reaction, while the intestinal phase was subjected to a boiling water bath for 5.0 min to end the reaction. The samples were centrifuged at 8000× *g* for 20 min at 4 °C. The supernatant and the sediment were separated and stored at −18 °C for the next analysis.

### 2.4. Degree of Hydrolysis

The OPA method developed by Zenker et al. [21] was used to determine the degree of hydrolysis (DH) of pork and grass carp subjected to IVD. To obtain the total content of free amino groups, 0.5 g of pork and grass carp were separately placed in 25 mL of 6.0 M HCl and hydrolyzed at 110 °C for 24 h. 10 μL diluted samples (20-fold by PBS, pH = 7.0) were then mixed with 200 μL of OPA reagent and incubated for 2 min, followed by measuring the absorbance at 340 nm using a microplate reader. A standard curve was prepared using L-leucine (0.2~1.0 mg/mL), and the DH of each sample was calculated using the following formula:(1)$$\mathrm{DH}=\frac{NH_{2}\;(\mathrm{final})}{NH_{2}\;(\mathrm{acid})}\times 100\%$$

Here, NH_2_ (final) refers to the concentration of free amino groups in the digested sample, and NH_2_ (acid) is the total content of free amino groups in the sample after acid hydrolysis.

### 2.5. Particle Size and Zeta Potential Determination

The supernatant samples were diluted to 1.0 mg/mL (pH = 7.0). According to the method proposed by Leng, et al. [22], the particle size and zeta potential of the soluble protein were determined using a Zetasizer Nano-ZS90 (Malvern Instruments, Malvern, UK). The refractive index used for the dispersed phase was 1.450, corresponding to a water RI of 1.333, at a reaction temperature of 25 °C and a scattering angle of 173°. The particle size distribution was intensity weighted.

### 2.6. Microscopic Morphological Observations

The microstructure of the digestions was observed by a transmission electron microscopy (HT7700, Hitachi Limited, Tokyo, Japan). Briefly, copper sieves (400 mesh) were immersed in the digestion solutions (0.1 mg/mL), then removed and dried at room temperature for 10 min. The morphology of the samples was observed by transmission electron microscopy at 80 kV acceleration voltage and 7000 magnifications.

### 2.7. SDS-PAGE

The protein concentrations of the supernatant samples were adjusted to 2.0 mg/mL with deionized water. Each sample was mixed with the loading buffer at a ratio of 4:1, then boiled at 95 °C for 5 min. The proteins in the digestion solution were separated and identified using 12% separator gel. Then, 10 μL of each lane was sampled and electrophoresed at 120 V, 25 °C for 1 h until the bromophenol blue indicator at the front of the band reached the bottom. After electrophoresis, the gels were stained with Thomas Brilliant Blue R-250 for 1 h, then decolorized until the bands were clear and the background was nearly colorless. The gels were finally scanned by a Gel Doc XR+ automatic gel imager (Bio-Rad, Hercules, CA, USA) and the image was saved.

### 2.8. TCA-Soluble Peptides

The protein concentrations of the supernatant samples were determined with the BCA kit and adjusted to 1.0 mg/mL with deionized water. The content of TCA-soluble peptides was measured according to the method by Buamard and Benjakul [23]. Briefly, 400 μL cold 5% TCA solution was added to 100 μL sample and the mixture was shaken well. The homogenate was stored in ice for 1 h and centrifuged at 8000× *g* for 10 min. The TCA-soluble peptides content in the supernatant was measured using the BCA kit and expressed as μmol tyrosine equivalent/mg protein.

### 2.9. Molecular Weight Distribution

The molecular weight distribution of peptide fractions in gastric and gastrointestinal digests of pork and grass carp were determined by GPC analysis using a 1260 Infinity II LC System (Agilent, Santa Clara, CA, USA) equipped with a G1362A refractive index detector and three serially connected Ultrahydrogel columns (TM120, TM250, and TM500, Waters, Milford, MA, USA). The column dimensions were 7.8 mm × 300 mm, and the mobile phase was a 0.1 M NaNO_3_ aqueous solution with a flow rate of 1 mL/min at 40 °C. After filtering the samples through a 10 kDa ultrafiltration membrane, 40 μL of each sample was injected and eluted for 35 min. The elution curve was recorded, and the molecular weight distribution was calculated based on the MW calibration curve. The MW calibration curve was generated using known retention times of the following standard proteins: vitamin B12 (6500 Da), glycylglycine (3157 Da), angiotensin II (1046 Da), oxidized glutathione (612 Da), and tyrosine (181 Da).

### 2.10. Amino Acid Composition

With reference to the method described by Lorieau, et al. [24], the composition of free amino acids and total amino acids in each digestive fluid sample were determined separately: 100 μL 10% sulfosalicylic acid solution was added to 400 μL of digestion solution and left at 4 °C for 1 h. After centrifugation at 8000× *g* for 15 min, the supernatant was taken and diluted 2-fold with buffer solution (pH = 2.2); the same fraction of digestion solution was hydrolyzed with 6 M HCl at 110 °C for 24 h. After drying in a rotary evaporator, the residue was suspended with 1 mL of buffer solution (pH = 2.2). The above samples were further analyzed by an automatic amino acid analyzer (L-8900 Hitachi, Tokyo, Japan). The difference between total and free amino acid content in the digestion solution was attributed to soluble proteins and peptides.

### 2.11. Circular Dichroism (CD)

The CD spectra of the soluble protein solution were measured using the JASCO J-1500 spectropolarimeter (Jasco Corporation, Tokyo, Japan). The samples were diluted to 0.1 mg/mL with PB buffer (0.01 M, pH = 7.0), filtered through a 0.45 μm membrane, and then loaded into a colorimetric dish with a light diameter of 1 mm. The scan speed, scan range, and spectral resolution were set to 100 nm/min, 190~250 nm, and 1 nm. The spectra were an average of 3 scans, and the secondary structure was calculated using Yang secondary structure model.

### 2.12. Fluorescence Spectroscopy Measurement

The protein concentration of the digestive fluids was adjusted to 1 mg/mL. Fluorescence spectra of the reaction solution were scanned at an excitation wavelength of 280 nm with a slit width of 5 nm and a scan rate of 240 nm/min. The scan was conducted within the wavelength range of 300~450 nm. Synchronous fluorescence spectra were also recorded for the sample in the wavelength range of 260~380 nm, with wavelength differences (Δλ) of 15 and 60 nm.

### 2.13. Antioxidant Capacity Assay

#### 2.13.1. ABTS^+^ Scavenging Activity

The ABTS^+^·scavenging activity of the supernatant samples were determined according to the method developed by Yang, et al. [25]. Each measurement was performed in triplicate to take the average, and the antioxidant capacities of the samples were expressed in Trolox equivalents.

#### 2.13.2. DPPH· Scavenging Activity

The method was modified from the former research in [26]. In total, 100 μL of supernatant samples (1.0 mg/mL) were mixed with 100 μL of 0.2 mmol/L DPPH (dissolved in 95% ethanol) as the sample group. The control group was 100 μL of deionized water mixed with 100 μL of 0.2 mmol/L DPPH, and the blank consisted of 100 μL ethanol (95%) and 100 μL samples (1.0 mg/mL). The mixtures were shaken and incubated for 30 min at room temperature without light, and then the absorbance at 517 nm was measured by using a microplate reader (Mutiskan SkyHigh, Thermo Fisher Scientific, Waltham, MA, USA). The DPPH· scavenging activity of the samples was calculated using the following formula:(2)$$\mathrm{DPPH}\cdot \;\mathrm{scavenging}\;\mathrm{activity}=(1-\frac{A_{s}-A_{b}}{A_{c}-A_{b}})\times 100\%$$
where *A_s_*, *A_b_*, and *A_c_* represent the absorbances of the sample, the blank, and the control groups, respectively.

#### 2.13.3. Ferric Reducing Ability

The T-AOC assay kit with FRAP method was used to determine the ferric reducing ability. In brief, a working solution of Tripyridyltriazine (TPTZ) was configured in a 10:1:1 ratio of TPTZ diluent, TPTZ solution, and assay buffer. Subsequently, 5 μL of samples (1 mg/mL) were added to the 180 μL working solutions and the absorbances of the mixtures at 593 nm were determined after 5 min of incubation at 37 °C. A standard curve was obtained using Trolox solutions with concentrations ranging from 0.15 to 1.5 μM, and the antioxidant capacity of the sample was expressed as Trolox equivalent.

#### 2.13.4. Fe^2+^ Chelating Ability

The Fe^2+^ chelating ability of the supernatant samples was evaluated using the modified method described by Yang, Huang, Dong, Zhang, Zhou, Huang, and Zhou [25]. The 160 μL samples (1 mg/mL) were mixed with 8 μL 2 mM FeCl_2_ and 32 μL 5 mM phenanthroline and incubated at room temperature for 10 min. Immediately after that, the absorbances of the mixtures were measured at 562 nm. The rest of the conditions remained unchanged, and the samples were replaced with deionized water as a control, and the Fe^2+^ chelating ability was calculated as follows:(3)$${\mathrm{Fe}}^{2+}\;\mathrm{chelating}\;\mathrm{ability}=\frac{A_{c}-A_{s}}{A_{c}}\times 100\%$$
where *A_s_* and *A_c_* represent the absorbance of the sample and control.

### 2.14. Statistical Analysis

All the tests were repeated in triplicate, and the results were expressed as mean ± SD (standard deviation). Data were evaluated using SPSS (SPSS Statistics 26, IBM, Amonk, NY, USA). Differences between groups were assessed using the independent sample *t*-test, and Duncan’s multiple range tests were employed to determine any significant differences within groups (*p* < 0.05). Correlation analysis was performed using the Pearson correlation test at https://www.omicstudio.cn/tool, accessed on 3 February 2023.

## 3. Results and Discussion

### 3.1. Degree of Hydrolysis

Degree of hydrolysis (DH) refers to the percentage of peptide bonds in a protein that are hydrolyzed. Figure 2 depicts the DH of pork and grass carp digested by pepsin and trypsin. The DH of both pork and grass carp protein significantly increased (*p* < 0.05) as digestion progressed, with a greater increase during the intestinal phase than during the gastric phase. This is because pepsin is an endopeptidase that primarily cleaves internal peptide bonds and can only produce larger peptides [27]; thus, its contribution to DH is far lower than that of trypsin. The number of peptide bonds hydrolyzed in grass carp exceeded that in pork during both gastric (14.06% vs. 10.46%) and intestinal (29.15% vs. 25.94%) digestion phases, which may indicate that grass carp released more peptides or amino acids throughout digestion and has better bioavailability. However, since DH measurement is based on free amino groups, it cannot effectively distinguish between peptide and amino acid release; therefore, further experiments are needed to confirm this. It is worth noting that, in this study, the degree of protein hydrolysis in pork after gastrointestinal digestion was lower than that reported by Wang et al. [28] and Martini et al. [29]. This difference may be attributed to the use of different enzymes, pancreatin and trypsin, during the intestinal digestion phase, resulting in variations in hydrolysis efficiency.

### 3.2. Particle Size Distribution and Zeta Potential

Particle size is an important physical parameter for characterizing the digestive process of food, and changes in particle size are directly related to the hydrolysis and aggregation of proteins during digestion [30,31]. Figure 3 shows the particle size distribution of protein particles in pork and grass carp after different digestion stages, with their mean particle size, PDI, and zeta potential illustrated in Table 1. It was found that the mean particle size of pork was significantly larger than that of grass carp at all stages of digestion (*p* < 0.05). For grass carp, the mean particle size decreased rapidly during the gastric digestion stage, and a bimodal size distribution was observed in the particle size distribution graph, with two populations around 70 nm and 1300 nm, respectively. The first aggregate indicates the formation of a higher level of small particles (30~110 nm) [31], and the polymer dispersibility index (PDI) was at a maximum value (0.53) throughout the digestion stage, indicating a wider molecular weight distribution [32]. In contrast, the particle size distribution of pork was more uniform during the gastric digestion phase, which had a narrow size distribution (PDI < 0.4) [33], and the bimodal distribution was not observed until the intestinal stage. As might be expected, the dispersion of the digested products was always broader compared with the undigested ones for both pork and grass carp.

Zeta potential is a key physical parameter that characterizes the surface charge of proteins. The larger the absolute value, the smaller the dispersed particles and the more stable the system. Values between −30 mV and +30 mV usually indicate instability or aggregation [34]. The absolute values of the zeta potential for both meats to digest increased gradually with the digestion process. This is because, before protease digestion, there are fewer like charges carried by proteins in the digestive fluid. The reduction of electrostatic repulsion reduces the stability of the solution and protein molecules tend to aggregate [35]. After hydrolysis into peptides and amino acids, the number of like charges increases, and the resulting repulsion makes the protein solution more stable. At the same time, aggregation between protein molecules also weakens. There is no significant difference in zeta potential between PID and GID after intestinal digestion (*p* > 0.05), and negative zeta potential values (pH = 7.0) were observed in both digestive fluids after gastric and intestinal digestion, suggesting that the amino acids or peptides released during their digestion phase may be more acidic.

### 3.3. Microstructure

The microstructures of both meat digests were observed by TEM (Figure 4), and the white particulate matter was identified as protein particles suspended in solution [36]. Throughout the digestion process, the pork protein granules were round in shape, with particle diameters varying significantly from about 1 μm before digestion to 0.2~0.3 μm after gastrointestinal digestion, whereas the GBD and GGD particles were mostly irregular in shape and a clear desire for separation could be observed. The GBD particles varied considerably in size, with equivalent particle sizes ranging from approximately 0.3 to 0.6 μm, whereas the GID after gastrointestinal digestion was essentially the same shape as the PID, with slightly smaller particle sizes ranging from approximately 0.1 to 0.2 μm. With each stage of digestion, the particle size of the protein significantly decreased. This phenomenon was also observed by Chen et al. in their study on the digestion of wheat germ protein [37]. The results of the trend observed by TEM are in general agreement with the conclusions drawn from the determination of the particle size distribution but differ in terms of the specific values. This may, on the one hand, be due to the higher protein concentration during the particle size determination, which led to some aggregation of the particles in the water, thus making the results greater than those observed directly with the TEM; on the other hand, the two experimental means present results from different angles, hence there is no direct agreement between the two.

### 3.4. SDS-PAGE

SDS-PAGE analysis revealed the molecular weight distribution of the digestates of pork and grass carp meat. The pre-digestion samples showed typical protein bands, such as the bands at 48 kDa (β-enolase), 42 kDa (actin), and 18 kDa (myoglobin) in pork [38], and the bands at 38 kDa (tropomyosin), 21 kDa (troponin), and 10~17 kDa (myosin light chain) in grass carp meat [39] (Figure 5). MHC (220 kDa) is a typical protein in meat, but it is difficult to observe on the bands of PBD and GBD. This may be related to its low solubility and short oral digestion time. Even with a simulated chewing behavior using a grinding treatment, it cannot be well dissolved in digestive fluid. Similar observations have been made in previous studies [40,41,42]. A slight shift in the bands above 250 kDa was observed in PGD compared with PBD, indicating that these proteins underwent some hydrolysis during gastric digestion resulting in smaller molecular weights; in the GGD, there was no longer a protein above 250 kDa. The GGD showed a new dark band at around 50 kDa, which could be from the hydrolysis of high molecular weight proteins, such as myosin heavy chains or myosin aggregates [43], or it could be due to the production of soluble fragments from some insoluble proteins as digestion proceeded [44]. These results suggest that there are differences in the hydrolytic properties of pork and grass carp meat proteins during the gastric digestion phase. After completing gastric and intestinal digestion, however, the electrophoretic profiles of the two digests were no longer significantly different: no clear bands could be observed in either and the bands were diffusely distributed.

### 3.5. TCA-Soluble Peptides

The TCA-soluble peptides content reflects the presence of oligopeptides in the digestate [45]. Given that lower molecular weight peptides tend to be more bioactive [46], this measurement of TCA-soluble peptides in the digestate can offer insight into the bioavailability and bioactivity potential of the proteins. Both pork and grass carp released a large number of TCA-soluble peptides during gastric digestion (Figure 6), with grass carp being significantly higher than pork, indicating that more oligopeptides were released. The increase in TCA-soluble peptides during the intestinal phase was less than that in the gastric phase, which was due to the release of trypsin mainly to amino acids at the end of the peptide chain.

### 3.6. Molecular Weight Distribution

After simulated digestion, pork and grass carp proteins were hydrolyzed into peptides and amino acids in large quantities. However, the molecular weight distribution of peptides was not well characterized in SDS-PAGE. Therefore, we used the GPC method to determine the molecular weight distribution of peptide components obtained from pork and grass carp after gastric digestion and gastrointestinal digestion, and the chromatograms are shown in Figure 7a–d. According to the chromatograms and the standard curve (lgMw = −0.1591t + 7.3671), 1000 Da was used as the scale for dividing the range and plotting the percentage of molecular weight distribution of the samples (Figure 7e). As can be seen from the figure, after gastric digestion, the peptides with the highest proportion in PGD and GGD have MWs between 1000~2000 Da. Peptides with MWs less than 3000 Da account for more than 50% (58% and 66%), indicating that gastric digestion releases a large number of small peptides from pork and grass carp meat. This is consistent with the conclusion drawn in Section 3.5. However, compared with GGD, peptide distribution below 4000 Da in PGD is more uniform with each range accounting for about 20%. The peptides with the least proportion are also different between PGD and GGD: only 8% of peptides are above 5000 Da in GGD while this ratio is 14% in PGD, indicating that there are still more polypeptides and proteins to be digested in PGD.

After intestinal digestion, the predominant peptide range in PID and GID shifts to below 1000 Da, accounting for more than 30% (36% and 46%), although this may include some free amino acids. Peptides and amino acids within this molecular weight range contributed the most to the degree of hydrolysis [47], and this result is consistent with the conclusion that enteral digestion results in a higher degree of hydrolysis, as discussed in Section 3.1. Compared with pre-intestinal digestion, the proportion of peptides above 2000 Da in both PID and GID decreased, and the difference in their total proportion was only 3% (29% vs. 26%), while this difference was 10% in the gastric phase. This may indicate that the species differences between pork and grass carp have less effect on trypsin hydrolysis than pepsin for peptides above 2000 Da. Similar phenomena were also observed in the results of Section 3.4, where there was no significant difference in SDS-PAGE bands between pork and grass carp after gastrointestinal digestion. Another interesting phenomenon is that the proportion of peptides in GID between 1000~2000 Da remains relatively stable. This may be because, within this range, peptides undergo continuous hydrolysis into smaller peptides, while higher molecular weight peptides are also hydrolyzed and replenished into this range, resulting in a dynamic equilibrium.

### 3.7. Free Amino Acids

The content and composition of free amino acids in digestive juices is an important indicator to evaluate the digestive characteristics and nutritional value of food [48]. As shown in Table 2, the amino acids of pork and grass carp meat were mainly released during intestinal digestion, with 25.73% and 37.32% of the total nitrogen content in PID and GID, respectively. The EAA/TAA values of both meats increased as digestion proceeded and exceeded 40% at the end of the intestinal phase, which is fully compliant with the FAO/WHO ideal protein standard of 40%. This means that intestinal digestion can effectively facilitate the release of nutrients from both types of meat. For the pre-digestion phase, the total amount of free amino acids was higher in the PBD than in the GBD (*p* < 0.05), which probably originated from thermal degradation and physical fragmentation of the meat during the cooking process. After gastric digestion, PGD was rich in Gln, Ala, and Phe, while GGD contained mainly His, Gly, and Pro, with Ala, Phe, and Pro being hydrophobic amino acids. This may be due to the opening of the hydrophobic structure inside the insoluble protein during gastric digestion, resulting in the production of more soluble proteins and hydrophobic amino acids. After intestinal digestion, there was no significant difference in the content of some free amino acids between PID and GID (*p* > 0.05), and the differences in the remaining amino acids may be attributed to variations in the sources of the proteins [49]. PID was rich in Leu, Phe, and Arg while GID was rich in Lys, Phe, and Arg. This suggested that, for grass carp meat, the basic amino acids (Lys, Arg) are more readily released during intestinal digestion, which is consistent with previous reports [50]; both were rich in Phe and Arg, probably due to the peptide bond formed by trypsin acting as a peptide chain endonuclease that selectively acts on Phe and Arg residues.

### 3.8. Amino Acid Composition of the Peptides

The amino acid composition of the peptide fraction in the digestate can be obtained by determining the total content of each amino acid and subtracting the content of the corresponding free amino acids [51]. The antioxidant activity of peptides is closely related to the composition, arrangement, and hydrophobicity of the amino acids in their sequence [52]. Studies have shown that peptides with a higher proportion of hydrophobic amino acids (Ala, Phe, Pro, Val, Leu, Ile, Met) often show better antioxidant activity [53], probably because the hydrophobic side chains of amino acids enhance the interaction between the peptide and oxygen radicals, thus effectively scavenging oxygen radicals and acting as an antioxidant [54]. During gastric digestion, PGD and GGD contained 31.48% and 33.56% of hydrophobic amino acids, respectively, decreasing to 22.34% and 25.15% during intestinal digestion (Figure 8). This may result in some of the antioxidant peptides produced after gastric digestion being inactivated by hydrolysis during intestinal digestion.

The acidity and basicity of amino acids also affect the antioxidant activity of peptides. The acidic amino acid (Asp, Glu) side chain carboxyl groups in the antioxidant peptides were reported to blunt the metal ions and weaken the free radical chain reaction, thus achieving the antioxidant effect [55]. The acidic amino acid content of the peptides in GGD (29.33%) and GID (47.16%) was higher than that in PGD (25.95%) and PID (36.1%), and the hydrophobic amino acid content showed the same trend. This difference may be partly due to the inherent differences in amino acid composition between the two types of muscle.

### 3.9. Circular Dichroism

The secondary structure of protein is the combination of different conformations in polypeptide chains, which can be analyzed quickly by CD spectroscopy. As indicated in Figure 9, before gastric digestion, the main secondary structures in PBD were the α-helix and β-sheet, the sum of which was 67.1%, higher than that of GBD (58.9%). The ordered rigid structure of polypeptide can be attributed to α-helix and β-sheet with more hydrogen bonds [56], which led to the conclusion that PBD has better stability than GBD before digestion. After gastric digestion, the α-helix structure of PGD and GGD disappeared, which may indicate a significant change in the secondary structure of actomyosin [57]. The proportion of β-sheets decreased while the proportion of random coil exceeded 50%, resulting in a significant increase in the disorder of the peptide chain [58]. Previous reports have indicated that the secondary structure of bioactive peptides is mainly dominated by β-sheet and random coil, with a higher proportion of random coil contributing to greater antioxidant activity [59]. According to the study conducted by Tang and Sun [60], protein molecules with a higher content of random coil and a lower content of β-sheet exhibit less compact structures and are more susceptible to hydrolysis.

After intestinal digestion, the secondary structure of both meats did not change much. This result may be due to the better digestive resistance of the β-sheet compared with the α-helix, or to the fact that the trypsin cleavage site mainly targets the endpoints of the peptide chain and has less effect on the secondary structure of the peptide chain. Throughout the digestion process, the secondary structure of all pork types changed more than that of grass carp meat, implying that digestion had a greater impact on the secondary structure of pork proteins.

### 3.10. Fluorescence Spectroscopy Analysis

To investigate the tertiary structure changes of proteins during digestion, the intrinsic and synchronous fluorescence spectra of pork and grass carp proteins at each digestion stage were analyzed (Figure 10). Most proteins have fluorescence characteristics due to the existence of aromatic amino acids (Trp, Tyr, and Phe), and the changes in the microenvironment of these three amino acids can be observed through the variation in protein intrinsic fluorescence. As shown in Figure 10a, the maximum emission wavelength of PBD and GBD is beyond 330 nm (333 nm and 335 nm), indicating a relatively polar microenvironment [61]. During gastrointestinal digestion, the fluorescence intensities of both proteins increased and the maximum emission wavelength redshifted by 11 nm and 13 nm, respectively. These changes are consistent with previous observations of myoglobin digestion by Li et al. [62]. It is possible that the shielding of some fluorescent amino acid residues by protein aggregation before gastric proteolysis led to weaker fluorescence intensities [63]. As the digestion proceeded and the protein spatial structure unfolded, the shielding effect was relieved, resulting in a stronger fluorescence intensity and more exposed chromophores in a polar environment. This is similar to the fluorescence changes observed during protein high-pressure denaturation [64], although the underlying mechanisms may differ. During the digestion process, a greater fluorescence intensity change was observed in pork protein compared with grass carp, suggesting that the tertiary structure of pork protein is more susceptible to the impact of digestion [62]. These changes in tertiary structure may increase the likelihood of enzymatic hydrolysis [65].

Synchronous fluorescence spectroscopy can obtain single characteristic fluorescence peaks of specific amino acids, allowing for a more precise analysis of tertiary structure changes in proteins. When Δλ = 15 nm, only the fluorescence spectrum of Tyr residues is displayed. As shown in Figure 10b, the fluorescence intensity of Tyr residues in PGD and GGD significantly increased after enzymatic hydrolysis, accompanied by a slight blueshift (3 nm and 2 nm). This may be due to the exposure of buried hydrophobic groups within the protein, leading to enhanced local hydrophobicity of the Tyr residues’ microenvironment [66]. After both gastric and gastrointestinal digestion, the two types of meat proteins exhibited almost identical characteristic fluorescence spectra, indicating that digestion does not have a specific differential effect on the Tyr residues’ microenvironment of the two proteins. At Δλ = 60 nm, only the fluorescence spectral characteristics of Trp residues were visible. Figure 10c shows that the characteristic spectrum of Trp residues changed similarly to the intrinsic spectrum, with an increase in fluorescence intensity and a red shift (3 nm). This is because Trp is the primary contributor to the protein’s fluorescence emission [67]. Therefore, some studies directly regard the intrinsic fluorescence spectrum of proteins as the spectrum of Trp residues [65,68].

### 3.11. Antioxidation

Based on previous research, it has been shown that meat protein sequences contain some bioactive sequences that exhibit functional properties in the form of bioactive peptides, such as antioxidant peptides, after hydrolysis by enzymes [69]. However, the enzymatic hydrolysis required for the release and activation of these peptides may also result in the loss of key antioxidant amino acids. Additionally, some insoluble proteins may also have antioxidant activity; although they cannot be absorbed in the small intestine, they may still be utilized by the colonic microbiota and exert an effect [70]. Therefore, it is important to note that IVD significantly affects the antioxidant capacity of the meat samples.

The antioxidant capacity of the digestive juices of pork and grass carp was measured at each stage from four separate angles: ABTS^+^· scavenging activity, DPPH· scavenging activity, FRAP (Ferric reducing ability), and Fe^2+^ chelating ability. As shown in Figure 11, both types of meat exhibit a certain level of antioxidant activity before gastric digestion, which may be attributed to the presence of some endogenous antioxidant peptides in meat, such as anserine and carnosine [71]. After gastric digestion, a significant increase in activity was observed in all four antioxidant indicators (*p* < 0.05), which can be attributed to the extensive release of specific antioxidant peptides. Zou et al. [42] identified four antioxidant peptides containing the known antioxidant sequences VDDLEGSLEQEKK and DDLEGSLEQEKK in simulated digestion products of pork. Martini, Conte, and Tagliazucchi [29] identified two known antioxidant peptides, VW and LW, in simulated digestion products of pork. These peptides likely contribute significantly to the antioxidant activity of pork digestive fluid. However, only the Fe^2+^ chelating ability showed an increasing trend after intestinal digestion (Figure 11d), possibly due to the hydrolysis of enzymes leading to an increase in the concentration of free amino and carboxyl groups in the digestive fluid, thereby enhancing the chelation of metal ions [72]. In addition, the Fe^2+^ chelating ability of grass carp was higher than that of pork after both gastric and intestinal digestion. This may be due to the higher degree of hydrolysis in grass carp, while in pork, the higher content of myoglobin, which is rich in iron ions, may reduce its chelating ability [73].

According to Figure 11a, there was no significant difference (*p* > 0.05) in ABTS^+^·scavenging activity between pork and grass carp digesta throughout the digestion process, although a slight decrease was observed after intestinal digestion, which was much smaller compared with the decreases observed in DPPH· scavenging activity and FRAP (Figure 11b,c). This may be due to the fact that ABTS is more sensitive to peptides containing Tyr residues [74] and, based on the results of amino acid analysis in Section 3.8, there was a slight reduction in Tyr content during intestinal digestion for both meats, which is consistent with the trend of changes in ABTS^+^· scavenging activity. A significant decrease in DPPH· scavenging activity and FRAP was observed after intestinal digestion for both meats, indicating a negative impact of excessive enzymatic hydrolysis on these two antioxidant abilities (Figure 11b,c). According to previous reports, it is known that after intestinal digestion, the digestive fluid contains a large number of short peptides and free amino acids, which have polarity and make it difficult to scavenge lipophilic DPPH [75]. Both FRAP and ABTS methods belong to total antioxidant capacity assays, but in this study, the antioxidant activity measured by the FRAP method was much lower than that measured by the ABTS method, which may indicate that the reducing ability of meat antioxidant peptides on trivalent iron ions is relatively poor [70]. Overall, neither of the meats showed an absolute advantage in antioxidant activity among the four measured antioxidant assays in all digestive stages.

### 3.12. Correlation Analysis

The antioxidant activity of digestion solutions may be influenced by multiple factors during the digestion process. In this study, five digestion-related indicators, including the degree of hydrolysis, particle size, TCA-soluble peptides, hydrophobic amino acid content, and random coil content, were selected and their correlation with the four antioxidant activity indicators for the two types of meat were established using Pearson correlation analysis (Figure 12). The correlation heatmap showed a strong correlation between the antioxidant activity of the two types of meat and their digestive characteristics. Specifically, the Fe^2+^ chelating ability of both types of meat was strongly correlated with DH, particle size, and TCA-soluble peptides, while their ABTS^+^· scavenging activity was closely related to random coil content, TCA-soluble peptides, and particle size. Their FRAP was influenced by random coil content, TCA-soluble peptides, and particle size. Therefore, it can be seen that TCA-soluble peptides, random coil content, and particle size have the greatest impact on the antioxidant activity of the two types of meat, corresponding to the production of oligopeptides, changes in protein structure, and overall digestion process, respectively. The correlation between hydrophobic amino acid content and antioxidant activity was more random, and no commonality was observed in the two types of meat, indicating that it is influenced by species differences. It should be noted that the DPPH· scavenging activity of grass carp did not show a strong correlation with any digestive characteristics, possibly due to its association with specific peptide sequences.

### 3.13. Limitations and Perspective

Due to technical and conditional limitations, there are indeed certain limitations in this work at present, such as the lack of specific information on peptides. We will conduct further in-depth research on this in our follow-up work.

## 4. Conclusions

Using pork and grass carp as examples, this study primarily investigated whether there are any significant differences in digestibility, nutritional value, and functionality between lean red meat and white meat after fat removal, and also explored their interrelationships. The results indicate that grass carp was more readily digested than pork, as evidenced by its higher degree of hydrolysis, smaller protein particle size, and greater release of oligopeptides and amino acids. The pepsin in the stomach digests the α-helix structure of both meat proteins, while the secondary and tertiary structure of pork protein is more affected by gastrointestinal digestion. The composition of free amino acids released from both meats after gastrointestinal digestion met the standard for high quality protein, indicating their high nutritional value. The antioxidant properties of the digestive fluids of the two meats focus on different aspects, and TCA-soluble peptides, random coil content, and particle size have a great impact on both of their antioxidant activity. By studying the digestive, nutritional, and functional characteristics of these typical meats, we can better understand their differences in digestion behavior and nutritional functional value, and provide some possible reference for people’s daily dietary choices.

## Figures and Tables

**Figure 1 foods-12-01757-f001:**
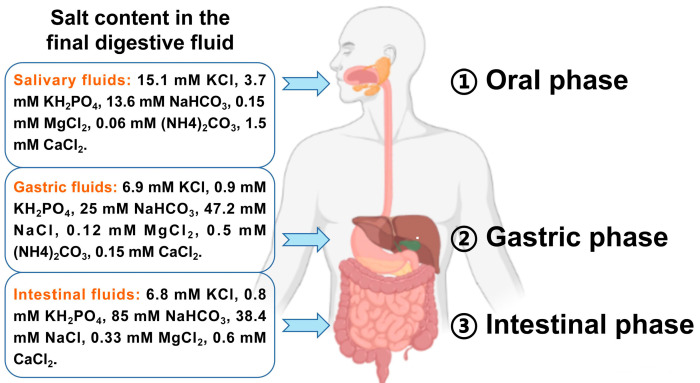
The digestion process and formulation of salivary, gastric, and intestinal fluids.

**Figure 2 foods-12-01757-f002:**
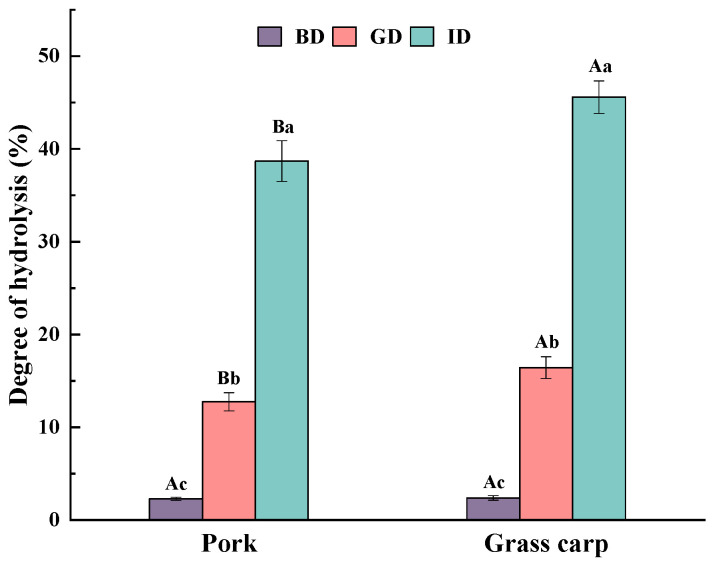
Degree of hydrolysis. BD, before digestion; GD, after gastric digestion; ID, after intestinal digestion. Different letters represent significant differences in DH for different types of meat (capital letters) and different stages of digestion (lowercase letters) (*p* < 0.05).

**Figure 3 foods-12-01757-f003:**
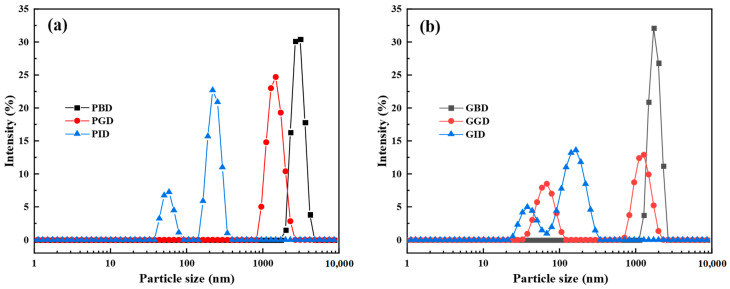
(**a**) Particle size distribution of pork before digestion (PBD), after gastric digestion (PGD), and after intestinal digestion (PID); (**b**) Particle size distribution of grass carp before digestion (GBD), after gastric digestion (GGD), and after intestinal digestion (GID).

**Figure 4 foods-12-01757-f004:**
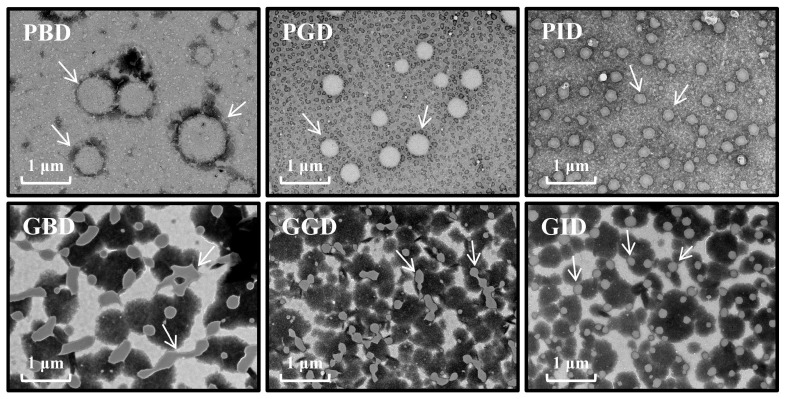
TEM images of the digestive juices of pork and grass carp meat at different stages of digestion. The white arrow indicates protein granules. PBD, pork before digestion; PGD, pork after gastric digestion; PID, pork after intestinal digestion; GBD, grass carp before digestion; GGD, grass carp after gastric digestion; GID, grass carp after intestinal digestion.

**Figure 5 foods-12-01757-f005:**
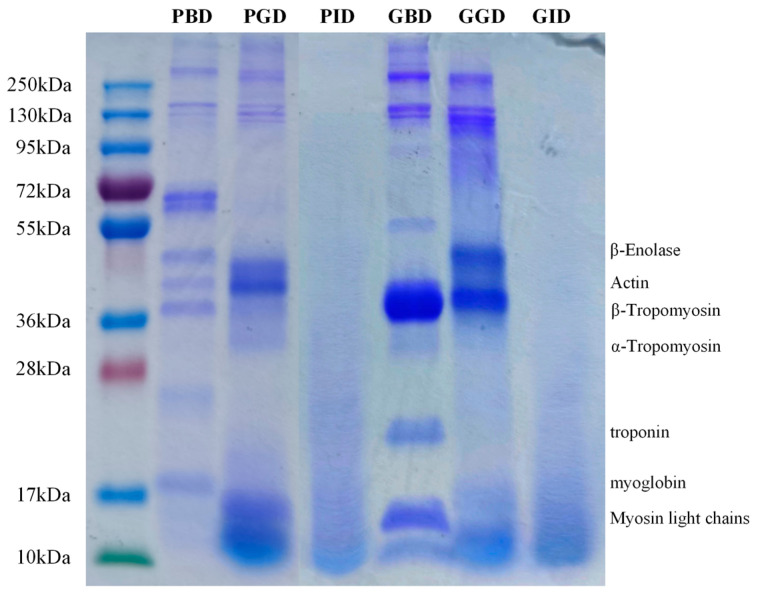
SDS-PAGE profiles of pork and grass carp meat at different stages of digestion. PBD, pork before digestion; PGD, pork after gastric digestion; PID, pork after intestinal digestion; GBD, grass carp before digestion; GGD, grass carp after gastric digestion; GID, grass carp after intestinal digestion.

**Figure 6 foods-12-01757-f006:**
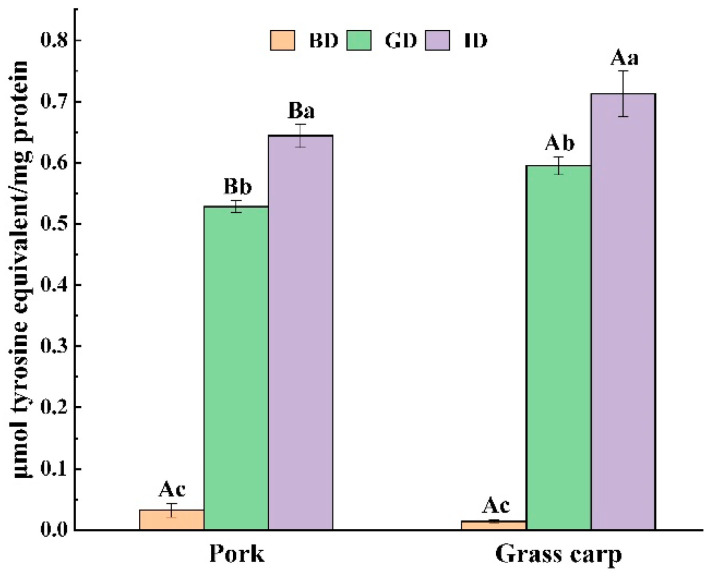
Changes in the TCA-soluble peptides content of pork and grass carp meat during digestion. BD, before digestion; GD, after gastric digestion; ID, after intestinal digestion. Different letters represent significant differences in TCA-soluble peptides content for different types of meat (capital letters) and different stages of digestion (lowercase letters) (*p* < 0.05).

**Figure 7 foods-12-01757-f007:**
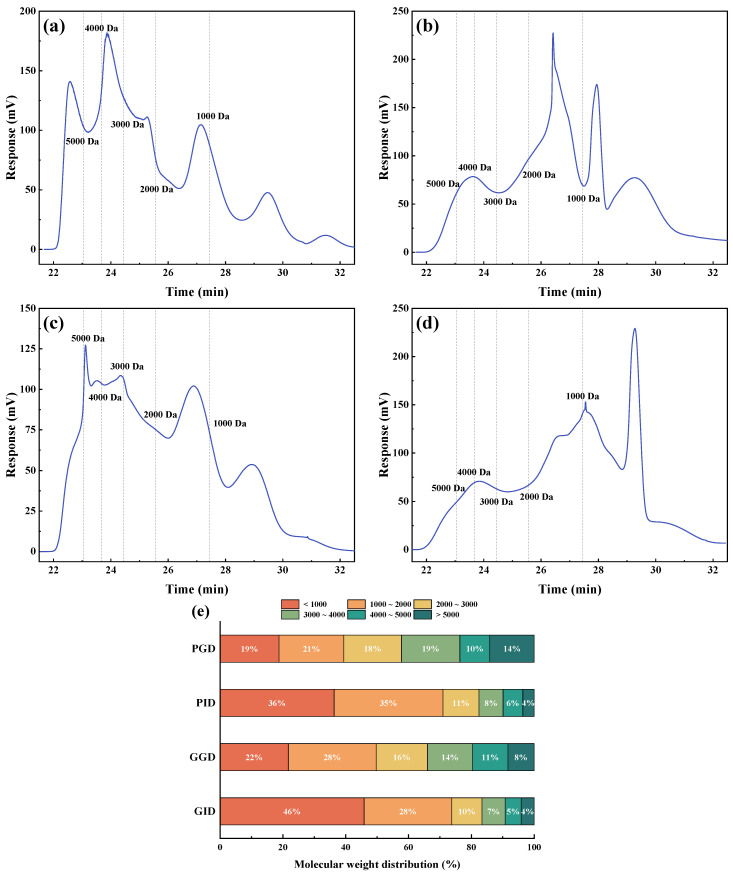
GPC chromatogram of (**a**) PGD, (**b**) PID, (**c**) GGD, and (**d**) GID; (**e**) the percentage of molecular weight distribution of peptides in the digestates of pork and grass carp at the end of gastric and gastrointestinal digestion. PBD, pork before digestion; PGD, pork after gastric digestion; PID, pork after intestinal digestion; GBD, grass carp before digestion; GGD, grass carp after gastric digestion; GID, grass carp after intestinal digestion. The location of the dotted line divides the range per 1000 kDa.

**Figure 8 foods-12-01757-f008:**
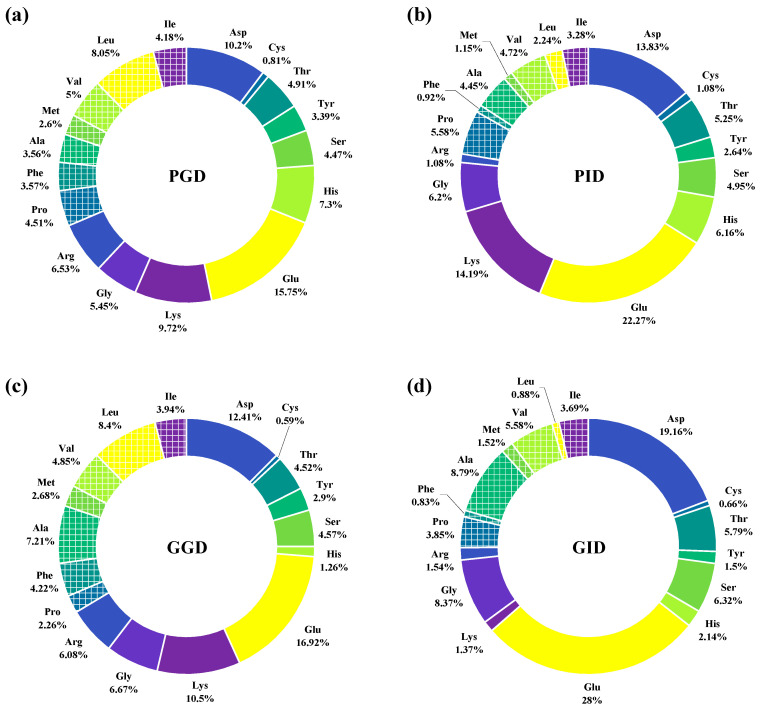
Amino acid composition of peptides in the digestive juices of pork and grass carp at the end of gastric and gastrointestinal digestion. (**a**): pork after gastric digestion; (**b**): pork after intestinal digestion; (**c**): grass carp after gastric digestion; (**d**): grass carp after intestinal digestion.

**Figure 9 foods-12-01757-f009:**
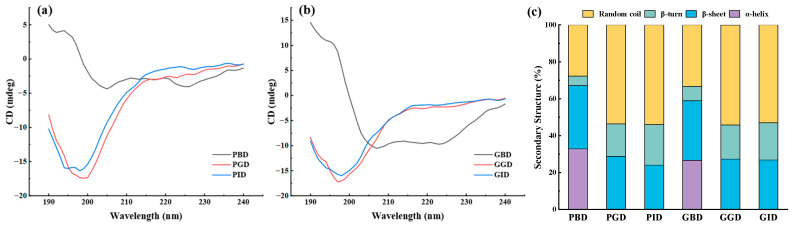
(**a**) Circular dichroism spectrum of pork at various stages of digestion; (**b**) Circular dichroism spectrum of grass carp at various stages of digestion; (**c**) Secondary structure content in the digestive juices of pork and grass carp meat at different stages. PBD, pork before digestion; PGD, pork after gastric digestion; PID, pork after intestinal digestion; GBD, grass carp before digestion; GGD, grass carp after gastric digestion; GID, grass carp after intestinal digestion.

**Figure 10 foods-12-01757-f010:**
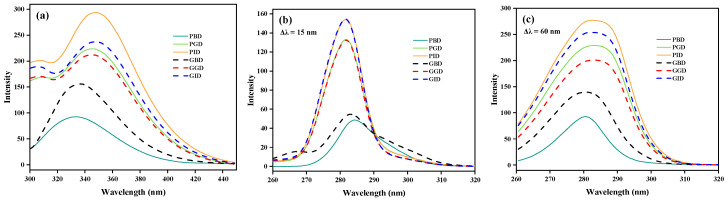
Fluorescence emission spectra (**a**), synchronous fluorescence spectra with Δλ = 15 nm (**b**), and Δλ = 60 nm (**c**) of pork and grass carp proteins at various stages of digestion. PBD, pork before digestion; PGD, pork after gastric digestion; PID, pork after intestinal digestion; GBD, grass carp before digestion; GGD, grass carp after gastric digestion; GID, grass carp after intestinal digestion.

**Figure 11 foods-12-01757-f011:**
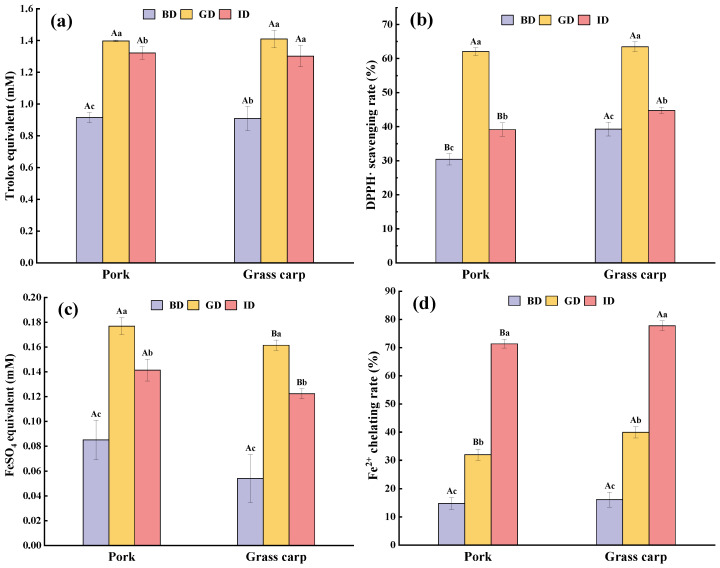
ABTS^+^· scavenging activity (**a**), DPPH· scavenging activity (**b**), Ferric reducing ability (**c**), and Fe^2+^ chelating ability (**d**) of the digestive juices of pork and grass carp meat at various stages of digestion. BD, before digestion; GD, after gastric digestion; ID, after intestinal digestion. Differences in capital and lowercase letters represent significant differences in the data for different types of meat and different stages of digestion, respectively (*p* < 0.05).

**Figure 12 foods-12-01757-f012:**
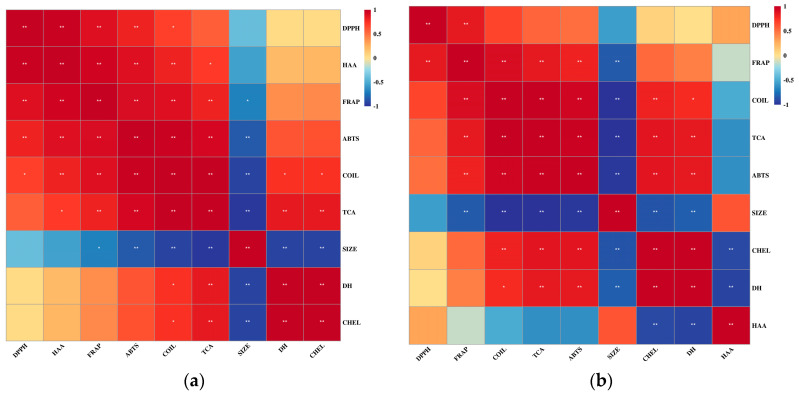
Heatmap of correlation analysis between different indicators for pork (**a**) and grass carp (**b**). Red represents positive correlation, blue represents negative correlation, and the intensity of the color reflects the size of the correlation coefficient. DPPH, DPPH· scavenging activity; HAA, hydrophobic amino acid; FRAP, Ferric reducing ability; ABTS, ABTS^+^· scavenging activity; COIL, random coil; TCA, TCA-soluble peptides; SIZE, particle size; DH, degree of hydrolysis; CHEL, Fe^2+^ chelating ability. The *p* value is marked with an asterisk (*) as *p* < 0.05, and two asterisks (**) are marked as *p* < 0.01.

**Table 1 foods-12-01757-t001:** Mean particle size, PDI, and zeta potential of pork and grass carp meat at various stages of digestion. PBD, pork before digestion; PGD, pork after gastric digestion; PID, pork after intestinal digestion; GBD, grass carp before digestion; GGD, grass carp after gastric digestion; GID, grass carp after intestinal digestion. Differences in capital and lowercase letters represent significant differences in the data for different types of meat and different stages of digestion, respectively (*p* < 0.05).

Samples	Mean Particle Size (nm)	PDI	Zeta Potential (mV)
PBD	2448 ± 141.45 ^Aa^	0.17 ± 0.06 ^Bc^	0.55 ± 0.01 ^Bc^
PGD	1197 ± 65.5 ^Ab^	0.27 ± 0.02 ^Bb^	−1.77 ± 0.03 ^Bb^
PID	459.5 ± 13.23 ^Ac^	0.55 ± 0.06 ^Aa^	−30.53 ± 0.12 ^Aa^
GBD	2074.7 ± 89.5 ^Ba^	0.20 ± 0.02 ^Ac^	−0.65 ± 0.03 ^Ac^
GGD	395.1 ± 60.42 ^Bb^	0.53 ± 0.05 ^Aa^	−2.73 ± 0.13 ^Ab^
GID	105 ± 0.87 ^Bc^	0.40 ± 0.01 ^Bb^	−30.33 ± 0.67 ^Aa^

**Table 2 foods-12-01757-t002:** Free amino acid composition of the digestive juices of pork and grass carp meat at different stages of digestion. PBD, pork before digestion; PGD, pork after gastric digestion; PID, pork after intestinal digestion; GBD, grass carp before digestion; GGD, grass carp after gastric digestion; GID, grass carp after intestinal digestion. Different letters represent significant differences in free amino contents for different types of meat (capital letters) and different stages of digestion (lowercase letters) (*p* < 0.05).

Amino Acid Content (mg/g Protein)	PBD	PGD	PID	GBD	GGD	GID
Asp	0.07 ± 0 ^Ab^	0.15 ± 0 ^Ab^	2.59 ± 0.06 ^Aa^	0.05 ± 0.01 ^Bb^	0.11 ± 0.02 ^Ab^	3.56 ± 1.88 ^Aa^
Glu	0.61 ± 0.02 ^Ac^	0.92 ± 0.06 ^Ab^	8.82 ± 0.26 ^Aa^	0.2 ± 0.03 ^Bb^	1.43 ± 0.26 ^Ab^	11.73 ± 1.9 ^Aa^
Asn	0.16 ± 0 ^Ab^	0.21 ± 0.02 ^Ab^	8.79 ± 0.39 ^Aa^	0 ± 0 ^Bb^	0 ± 0 ^Bb^	10.1 ± 1.83 ^Aa^
Ser	0.42 ± 0.09 ^Ab^	0.54 ± 0.02 ^Ab^	5.96 ± 0.38 ^Aa^	0.14 ± 0.02 ^Bb^	0.44 ± 0.04 ^Bb^	5.07 ± 0.8 ^Aa^
Gln	1.22 ± 0.1 ^Ab^	1.61 ± 0.12 ^Ab^	16.44 ± 1.59 ^Aa^	0.41 ± 0.02 ^Bb^	1.56 ± 0.07 ^Bb^	17.73 ± 3.26 ^Aa^
His	0.18 ± 0 ^Bb^	0.27 ± 0.05 ^Bb^	6.28 ± 0.26 ^Ba^	5.71 ± 0.31 ^Ab^	17.29 ± 1.15 ^Ab^	17.75 ± 3.49 ^Aa^
Gly	0.48 ± 0.14 ^Bc^	0.65 ± 0.02 ^Bb^	3.45 ± 0.03 ^Ba^	1.41 ± 0.06 ^Ac^	4.21 ± 0.56 ^Ab^	5.5 ± 0.93 ^Aa^
Thr	0.38 ± 0.07 ^Ab^	0.46 ± 0.13 ^Bb^	7.23 ± 0.14 ^Aa^	0.24 ± 0.01 ^Bb^	0.76 ± 0.1 ^Ab^	7.2 ± 1.4 ^Aa^
Arg	0.3 ± 0.08 ^Ab^	0.46 ± 0.08 ^Bb^	48.07 ± 3.43 ^Aa^	0.32 ± 0.02 ^Ab^	1.01 ± 0.13 ^Ab^	59.66 ± 5.95 ^Aa^
Ala	11.96 ± 0.6 ^Ac^	15.28 ± 1.19 ^Ab^	21.01 ± 0.77 ^Aa^	0.5 ± 0.02 ^Bb^	1.56 ± 0.36 ^Bb^	10.35 ± 1.73 ^Ba^
Tyr	0.35 ± 0.09 ^Ab^	0.6 ± 0.03 ^Bb^	9.72 ± 0.41 ^Ba^	0.16 ± 0.01 ^Bb^	0.65 ± 0.1 ^Ab^	32.19 ± 5.86 ^Aa^
Cys	0.03 ± 0 ^Bc^	0.25 ± 0.02 ^Ab^	0.71 ± 0.01 ^Aa^	0.05 ± 0 ^Ab^	0.18 ± 0.01 ^Bb^	1.25 ± 0.23 ^Aa^
Val	0.32 ± 0.05 ^Ab^	0.46 ± 0.01 ^Ab^	9.07 ± 0.18 ^Aa^	0.16 ± 0.04 ^Bb^	0.48 ± 0.01 ^Ab^	8.32 ± 1.62 ^Aa^
Met	0.28 ± 0.02 ^Ac^	0.78 ± 0.03 ^Ab^	12.27 ± 0.26 ^Aa^	0.09 ± 0 ^Bb^	0.45 ± 0.08 ^Bb^	10.73 ± 2.14 ^Aa^
Trp	0.07 ± 0 ^Bb^	0.13 ± 0.03 ^Bb^	10.59 ± 0.22 ^Aa^	0.19 ± 0 ^Ab^	0.29 ± 0.01 ^Ab^	9.07 ± 1.92 ^Aa^
Phe	0.4 ± 0.05 ^Ac^	1.83 ± 0.04 ^Bb^	26.94 ± 2.05 ^Aa^	0.25 ± 0.01 ^Bb^	2 ± 0.14 ^Ab^	35.86 ± 8.42 ^Aa^
Ile	0.31 ± 0.02 ^Ab^	0.55 ± 0.01 ^Bb^	12.22 ± 0.28 ^Aa^	0.22 ± 0.03 ^Bb^	1.16 ± 0.06 ^Ab^	11.23 ± 2.42 ^Aa^
Leu	0.52 ± 0.07 ^Ab^	1.31 ± 0.06 ^Bb^	45.14 ± 1.6 ^Aa^	0.33 ± 0.02 ^Bb^	1.65 ± 0.21 ^Ab^	46.8 ± 4.84 ^Aa^
Lys	0.36 ± 0.1 ^Bc^	0.51 ± 0.04 ^Bb^	0.98 ± 0.04 ^Ba^	0.78 ± 0.07 ^Ab^	2.35 ± 0.12 ^Ab^	60.57 ± 8.26 ^Aa^
Pro	0.24 ± 0.05 ^Bb^	0.51 ± 0.91 ^Bb^	1.02 ± 0.06 ^Ba^	3.15 ± 0.25 ^Ab^	8.09 ± 0.23 ^Aa^	8.5 ± 1.87 ^Aa^
TAA	18.66 ± 0.02 ^Ac^	27.47 ± 0.1 ^Bb^	257.29 ± 0.78 ^Ba^	14.37 ± 0.01 ^Bc^	45.66 ± 0.07 ^Ab^	373.18 ± 15.67 ^Aa^
EAA/TAA	14.14%	21.91%	48.36%	15.75%	20.00%	50.86%

## Data Availability

Data are contained within the article.

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
