# Peer review of "Comparative Study on Hydrolysis, Physicochemical and Antioxidant Properties in Simulated Digestion System between Cooked Pork and Fish Meat"

_foods, 2023, doi:10.3390/foods12091757_

Round 1

Reviewer 1 Report

Chen et al studied the digestion of Fish in relation with pork and its releasing thier peptides during the gastric digestion. The point is worth of study, but I have some points to be fixed before going to the next step:

- Why you did compare between fish and pork? Is there any relation between both meat so we can understand it? Is it be better to compare among larg animals or among fish types? Please explain with details as this should be your main hypothisis.

- I do recommend that the author cite those two related articles and see the experiemnts they used to confirm thier hypothis:

Wen, S., Zhou, G., Song, S., Xu, X., Voglmeir, J., Liu, L., ... & Li, C. (2015). Discrimination of in vitro and in vivo digestion products of meat proteins from pork, beef, chicken, and fish. Proteomics, 15(21), 3688-3698.

Khalifa, I., Zhu, W., Nawaz, A., Li, K., & Li, C. (2021). Microencapsulated mulberry anthocyanins promote the in vitro-digestibility of whey proteins in glycated energy-ball models. Food Chemistry345, 128805.

Park, C. S., & Adeola, O. (2022). Digestibility of amino acids in fish meal and blood-derived protein sources fed to pigs. Animal Bioscience, 35(9), 1418-1425.

Jiang, S., Zhang, M., Liu, H., Li, Q., Xue, D., Nian, Y., ... & Li, C. (2022). Ultrasound treatment can increase digestibility of myofibrillar protein of pork with modified atmosphere packaging. Food Chemistry, 377, 131811.

Khalifa, I., Lorenzo, J. M., Bangar, S. P., Morsy, O. M., Nawaz, A., Walayat, N., & Sobhy, R. (2022). Effect of the non-covalent and covalent interactions between proteins and mono-or di-glucoside anthocyanins on β-lactoglobulin-digestibility. Food Hydrocolloids133, 107952.

- Preparation of in vitro simulated digestion: explain this section in detail and I prefer to draw a digram to show the process and the compostion of each step.

- Particle size and zeta potential determination, what is the referctive index was used to measure this paramter?

- make a space between each number and the units.

- why you measured the CD and did not considre the FTIR, how did you calculate the CD results and is there any turbidity affected the performing of the CD experiment?

- Table 1, it is not acceptable to have a PDI over than 0.5, for the GGD it should be repeated to get more acccurate data. and for all samples have zeta potential between -30 to +30 these results showed unstable values, how to explain it?

- Add more explination on the SDS figure so the common readers could understand it.

Author Response

请参阅附件。

Reviewer 2 Report

This manuscript deals with the comparison on digestion between fish and pork meat using INFOGEST method. In fact, it has been known for many decades that fish meat or protein is easily digested and recommended to consume worldwide. Thus, this work does not provide any new information on this aspect. Also, the technical quality for some parameters tested was poor and misleading. Overall, this work is lacking in novelty and scientific merit.

Specific comments

1.     For INFOGEST, only trypsin used for intestinal tract was not convincing. Pancreatin containing several enzymes should be used instead. The pancreatin more likely mimic the enzymes present in the intestinal tract rather than trypsin alone.

2.     During the digestion, authors should report the degree of hydrolysis instead of TCA soluble peptides. This will indicate how many peptide bonds in both samples were cleaved in INFOGEST model system.

3.     Authors used dry matter to represent the digested proteins. However, during the digestion, the neutralization from stomach condition to intestinal condition could led to the salt formation. Thus, the total solid did not contain only the digested proteins, but also salt. Authors must consider on this artifact of result obtained.

4.     In this study, the size of peptides was not determined properly. Gel filtration chromatography should be used to monitor size distribution. However, the authors used particle size analysis to measure the size of peptide particles. In fact, peptides are soluble in nature, especially after centrifugation. The use of particle size analyzer has been used for colloidal system, containing disperse particles in aqueous phase. Therefore, this is the flaw in term of analysis point of view.

5.     Why only secondary structure of samples was determined in digest? In fact, protein has the globular structure, especially at the head of MHC. Also, after the hydrolysis, how did the confirmation play a major role in absorption or antioxidant activity? This is not clearly discussed.

6.     From SDS-PAGE protein pattern, it is hard to see the MHC band. Authors mentioned only (pro-myosin (38 kDa). What was pro-myosin mean? Under electrophoresis, MHC was dissociated to monomer. It is generally found MHC at 200 kDa and actin at 45 kDa. However, it is hard to see those protein bands, which constitute as the major proteins in the muscle in the result (Figure 4)

7.     For the antioxidant activity, not only amino acid composition, but also the amino acid sequence, size of peptide, hydrophobicity, etc. have the impact on the activity. However, there was no report on the amino acid sequence of peptides in this study.

8.     TEM images did not provide the in-depth information or related with any parameters tested. What were those particles? Are they oil droplets or other residual connective tissue? It is not verified or proven clearly on these images.

Reviewer 3 Report

Abstract lines 11 and 12 are redundant sentences and can be removed. 
For  INFOGEST digestion protocol, the composition of SSF fluid should be mentioned
as well as gastric fluid (SGF).
lines 28 and 29, please mention the causes of red meat to be unhealthy such as the high content of myoglobin, saturated fats, etc. that causes lipid oxidation and instability, ...
line 33: the global statistics on grass carp production in China as well as world from the latest FAO report?
line 44: "generation of some negative states in the human body" not clear!!!

line 71: Does INFOGEST digestion protocol considers the role of trypsin for intestinal digestion only, and not considering chymotrypsin?

line 119: for CD measurements, what was the buffer used to prepare the analysis solution?

line 282: However the changes of the content of acidic aa in pork or fish can be influenced by the inherent composition of aa in the two different muscle (pork or fish), thus the differences might partly be due to the inherent differences of amino acid composition between the two types of muscle. Please changes this part suitably. 

Findings shown in Fig. 6 is interesting, conforming the loss of alpha-helical elements along with increasing of random structure.

Conclusion: What was the novelty of this work, presenting the usefulness of INFIGEST method? It is not clear to me.

Round 2

Reviewer 1 Report

I congratulate the authors for thier response.

Author Response

Thank you very much for the professional comments.

Reviewer 2 Report

The authors have added some important data such as size distribution as well as degree of hydrolysis. However,  authors must amend the manuscript as follows:

1.     Firstly, to make the title is more interesting since the former one presents what people have learnt or known for a long time. The new title should be as follows:

‘Comparative study on hydrolysis, and physicochemical and antioxidant properties in simulated digestion system between cooked pork and fish meat’.  

2.     Data on digestibility calculated in the present form must be removed from the manuscript since salt generated from pH adjustment could lead to the over-estimated value. It is not scientifically sound to follow the wrong previous procedure.

3.     For digestion, pancreatin must be used instead of trypsin alone. However, only trypsin was used. Please make a further discussion on this in the text regarding the real system since there are more proteases rather than only trypsin.

4.     Size distribution was conducted using gel filtration chromatography. Please provide the chromatograms of all the sample, in which the distribution of size must be well demonstrated.

5.     Size of particle (dispersed matters) was also reported. Were those particles filtrated before GPC analysis? Please clarify and discuss in the text. With higher DH and smaller size, the solution should be obtained instead of suspension, which cannot be further absorbed through the intestine.

6.     MHC, which constitutes around 6070%, should have the typical MW (200 kDa). After short cooking, it is hard to form polymerized proteins as appeared at 250. It is not logically sound. However, authors can compare the result with one paper, which showed the same result on cooked pork meat.

Zou et al. 2020. Structural changes and evolution of peptides during chill storage of pork. Frontiers in Nutrition. Doi: 10.33889/fnut.2020.00151

7.     Discussion of antioxidant activity must be extended and the above reference which showed the peptide sequence can be used for discussion on antioxidant activity. They conducted the similar work using the simulated gastrointestinal tract and determine the peptide sequence of varying peptides after digestion of cooked pork meat.  

8.     In conclusion, the following statement ‘By studying the digestive, nutritional, functional characteristics of these typical meats, we can better understand the impact of meat on human health and may develop strategies to improve its health benefits for human consumption, thereby increasing the health value of food.”, is not convincing. The sentence is overstated. Please remove and provide the solid finding from the study instead.

9.     For all figure legends and table footnotes, the abbreviations must be described clearly.

Reviewer 3 Report

Authors adequately revised manuscript according to comments 

Author Response

(The authors gave the same response as above.)
